# Fighting the Deadly Helminthiasis without Drug Resistance

**George F. W. Haenlein [1],* and Young W. Park [2],***

[1]  Department of Animal and Food Sciences, University of Delaware, Newark, DE 19717, USA
[2]  Georgia Small Ruminant Research & Extension Center, Fort Valley State University, Fort Valley, GA 31030, USA
*  Correspondence: ghaenlein@gmail.com (G.F.W.H.); parky@fvsu.edu (Y.W.P.);
   Tel.: +1-304-743-8909 (G.F.W.H.); +1-478-827-3089 (Y.W.P.)

**Abstract:** Helminthiasis is a very costly management problem in the sheep and goat industry, because the gastrointestinal parasites develop resistance against all chemical products that are discovered and produced by the pharmaceutical industry. The use of natural herbal contents of tannin as especially in Sericea Lespedeza (SL; *Lespedeza cuneate*) is very promising. Utilizing genetic differences in resistance among the different goat and sheep breeds is a promising alternative, with limited success to date. Totally eliminating the offending parasites from re-infesting by plowing under affected pastures for some seasons, or scheduling rotational pastures, or feeding fresh (grazed) or dried forms of the perennial warm-season legume sericea lespedeza to the infected sheep and goats, or using elevated housing with slatted floors are the most promising alternatives to the ancient tradition of herding and managing ruminants by transhumance. An elevated slatted floor housing is desirable, and deserves wider attention because of its potential in controlling helminthiasis. Slatted floors are already used in the sheep and goat industries in Sweden, Norway, Malaysia and Guatemala.

**Keywords:** helminthiasis; anthelmintic drugs; tannin; Sericea Lespedeza; resistance; breed resistance; vaccination; plowing; elevated slatted floor housing

## 1. Introduction

Helminthiasis is the invasion and destructive life of internal parasitic worms into the gastrointestinal tracts of domesticated animals, and even humans, often with serious, even deadly, consequences. Many different endoparasite species can be found in sheep and goats worldwide, and sheep and goats are more seriously afflicted than other farm animals [1]. Millions of dollars are lost in decreased milk and meat production, reproduction failure, treatment costs and deaths. The use of anthelminthic drugs has been the main avenue for control. However, their use is limited by the occurrence of resistance to drugs. As they affect the parasitic worms incompletely, those left over produce pharmaceutical drug resistance against further treatment with the same drug. The pharmaceutical recommendation is that one must change from one to another drug at the next treatment, which needs to occur before the next reproductive cycle of the principle worm species, or use several de-wormer drugs in combination [2]. Thus, the fighting of the helminthiatic conditions of endoparasitic worms is complicated by the development and existence of pharmaceutical resistance, specifically de-wormer drug resistance.

Helminthiasis is attributed to poor growth rates, reduced reproductive performances, increased mortality, and low-quality products from infected ruminant animals, which eventually increase production costs [3]. These parasitic problems are worse in tropical and subtropical regions because of the high prevalence of infected pastures due to favorable environmental conditions for parasites to survive in their free-living stage [4]. Historically, anthelmintic drug treatment has been the most

common control method against gastrointestinal nematodes (GIN) infection. However, there has been overuse and misuse of this approach for the last 4–5 decades, which has led to a worldwide increase in the prevalence of anthelmintic resistance among major nematode species in small ruminants [5–7]. Over 90% of goat farms had high levels of GIN resistance to ivermectin and albendazole in a report from Georgia [8]. More recently, Howell et al. (2008) [9] reported total anthelmintic failure (resistance to all available anthelmintics) on 17% of sheep and goat farms throughout the southeastern USA.

Considering these unfavorable conditions of internal parasites in small ruminant production, the development of alternative, reliable and sustainable methods for controlling the GIN are undeniably desirable. The objective of this article is to review the fighting of helminthiasis in ruminants, especially in goat and sheep production, using several antihelmintic methods, including the use of traditional antihelmintiic drugs, feeding ruminant animals with natural forages with antihelmintic properties, and slated floor housing.

## 2. Drugs

There are at least a dozen different anthelmintic drugs in three major categories available for use by veterinarians and farmers [10]. In the category of Benzimidazoles, there is Fenbendazole, then Albendazole and Oxydendazole. Among the nicotinic agonists used are Levamisole, Morantel and Pyrantel, and among the macrocyclic lactones are Ivermectin, Doramectin, Eprinomectin and Moxidectin. These are the current drug names, and they are marketed under different tradenames for the most popular ones—Benzimidasole, Levamisole and Ivomec—until new ones are developed because of the problem of resistance. Recent studies of existing pharmaceutical resistance in sheep (14 farms) and goats (20 farms) in Maryland, Virginia and Georgia, USA [10] showed in the period 2008–2009 that 80% of all farms showed resistance to Ivermectin, 95% to Benzimidazole, 40% to Moxidectin, 20% to Levamisole, and 10% to all the drugs. Since then, in the last 10 years, the percentages had increased to 100, 97, 83, 50 and 33%. Thus, the effectiveness of current de-wormer drugs is decreasing, and when it falls below 50% it is no longer useful. The general recommendation had been to rotate among different antihelmintic drugs, but new research [2] in New Zealand has shown that combining several drugs is more effective.

## 3. Tannin Contents

Goats are more sensitive to parasitic gastrointestinal nematodes and helminth infections than cattle or sheep [11]. In addition, goats are not grazers like sheep and cattle, but they are, by nature, browsers, consuming preferentially tree leaves, twigs, bark, vines and bushes, which can mean less exposure to parasites, and consequently lower intestinal worm egg numbers in affected goats. However, goats also often host different internal species of parasites than sheep or cattle, therefore medications and nutritional control programs need to be specific to goats and are not appropriate to transfer from sheep and cattle [10]. Furthermore, goats have a different nutrition from sheep and cattle [12] including tolerance of plants with tannin contents [13], while sheep and cattle avoid tannin, do not tolerate tannin-containing plants, and may have depressed efficiency of their dietary energy utilization in the presence of tannins [14]. Tree leaves and bark have tannin content and this makes goats nutritionally more similar to deer, and they are not "small cows", as has been a popular attitude among scientists when nutritional or medical programs are needed. A recent study with different breeds of goats [15] showed genetic differences in the tolerance or even the selection of tannin-containing plants. Damascus goats under freely grazing conditions consumed three times more the tannin-rich Pistacia lenticus plant (116.4 g/day tannin) than their herdmate Mamber goats or crossbred Alpine goats. They also, under grazing conditions, produced higher milk protein (3.37%) and milk fat (6.12%) compositions and especially, in terms of human health potential, more medium chain fatty acids, more monounsaturated fatty acids (MUFA), 20% richer in omega 3 fatty acids and lower in the omega 6/omega 3 ratio.

The use of herbal medicines containing phenolic compounds as part of tannin contents has had wide native use in some countries, like Brazil [16]. Recent studies with tannin extracts from Spigelia

anthelmia showed a 100% inhibition of motility with ovicidal and larvicidal effects on the dominant nematode Haemonchus contortus, thus promising efficacy in helminth control. Phytochemical studies identified the alkaloid spiganthine as the major component responsible for the nematicidal activity of the herbal extract.

One of the most extensive studies about tannin involved Sericea Lespedeza with goats was conducted at the Agricultural Research Station Fort Valley, Georgia, USA [17]. Lespedeza is a perennial warm-season, drought-resistant legume forage and popular with soil conservation programs, but not with farmers trying to use it as pasture for cattle and sheep, because the tannin content makes it less palatable, while goats actually like it. Growth trials with 40 intact Kiko X Spanish male goats, 6 months old [18], showed the Lespedeza hay-fed goats to have a significantly greater average daily weight gain of 104 g/day over the 98-day long trial period compared to 76 g/day for the Bermudagrass-fed goats. The difference in better weight gain was explained by the significantly greater daily dry matter intake from Lespedeza hay (4.94 kg/day/pen) compared to 3.25 kg for the Bermuda grass hay group. Rumen fluid analyses also indicated that the Lespedeza-fed goats had a more efficient protein metabolism than the Bermudagrass fed goats, higher rumen volatile fatty acids (VFA) concentrations and lower acetate:proprionate ratios. Finally, the Lespedeza hay-fed goats had significantly lower fecal egg counts always.

Additional trials with 30 growing male Spanish goats, 9 months old [19], using Sericea Lespedeza hay pellets as a 75% or 95% supplement showed after 11 weeks grazing that on slaughter and carcass analyses the Lespedeza goats had significantly lower fecal egg counts than the goats fed commercial pellets. They also had significantly fewer nematodes in the abomasum and small intestine. Compared to the growing goats fed the commercial supplement, the reduction in fecal egg counts in the growing Lespedeza goats was 95% and 84% for the two Lespedeza leaf meal supplement levels, 95% and 75% in the daily ration, respectively (Table 1), and especially for the main parasite Haemonchus contortus.

**Table 1.** Fecal egg count of growing goats grazing and supplemented with commercial pellets or 75% Lespedeza leaf meal or 95% Lespedeza leaf meal (adapted from [19]).

| | Fecal Egg Count (Eggs/Gram Feces) | | |
|:---:|:---:|:---:|:---:|
| **Days after Start** | **Commercial** | **75% Supplement** | **95% Supplement** |
| 0 | 0 | 0 | 0 |
| 14 | 500 | 600 | 500 |
| 21 | 650 | 700 | 550 |
| 28 | 700 | 650 | 500 |
| 35 | 1000 | 600 | 400 |
| 42 | 950 | 550 | 300 |
| 49 | 1100 | 500 | 250 |
| 56 | 1400 | 400 | 200 |
| 63 | 1500 | 300 | 100 |
| 70 | 1600 | 250 | 100 |
| 77 | 1600 | 200 | 100 |

In addition to the antiparasitic effect of lowering fecal egg counts in the experimental goats, Sericea-Lespedeza-supplemented feeding significantly enhanced animal performance, such as in final body weight, daily weight gain and post-slaughter shrinkage in chevon meats, as shown in Table 2 [19]. Initial live weights of the goats were similar, but final live weights and shrink weights following pre-slaughter feed withdrawal were higher ($p < 0.05$) for goats fed 95% SL leaf meal pellets compared with animals given 75% SL pellets (Table 2) [19]. The average weight gain per day was also revealed to be significantly higher ($p < 0.05$) for the 95% SL pellet-fed group compared to the 75% SL group. Even if the differences were not statistically significant, final and shrink weights and average daily gains of the control animals were lower than for the 95% SL pellet group and greater than for the 75% SL-fed animals. There was no significant difference between carcass weights of all three treatment groups.

**Table 2.** Average body weight, daily gain, and post-slaughter shrink weight in growing goats' grazing. perennial summer grass pasture supplemented with 95% sericea lespedeza (SL) leaf meal, 75% SL leaf meal, or commercial pellets (adapted from [19]).

| Pellet | Body Weights (kg) | | | | |
|---|---|---|---|---|---|
| | Initial Live wt | Final Live wt | Gain per Day (g) | Shrink wt | Carcass wt |
| Commercial | 20.4 ± 0.95 [a,*] | 26.1 ± 1.20 [a,b] | 77.2 ± 0.5 [a,b] | 24.9 ± 1.18 [a,b] | 16.2 ± 0.63 [a] |
| 75% SL leaf meal | 19.7 ± 0.85 [a] | 23.7 ± 1.07 [b] | 53.3 ± 9.43 [b] | 23.4 ± 1.06 [b] | 15.1 ± 0.56 [a] |
| 95% SL leaf meal | 21.2 ± 1.02 [a] | 29.6 ± 1.28 [a] | 102.0 ± 11.2 [a] | 27.8 ± 1.27 [a] | 16.8 ± 0.67 [a] |

[a,b] Means with different superscripts within a same column are different at * $p < 0.05$.

In addition to the problem with nematodes there can also be a threat from coccidia (*Eimeria sp.*) to the health and productivity from small ruminants, especially young animals, whose immune system has not matured. Under favorable conditions such as transportation stress, there can be an outbreak of coccidiosis in a flock, with serious symptoms of diarrhea, dehydration and death [20]. In two studies with 24 Kiko-cross bucks and with 20 Spanish bucks, 16 and 20 weeks old, respectively, diets were supplemented with 90% lespedeza leaf meal pellets. Fecal egg counts and fecal oocyst counts both decreased again significantly in both studies. Coproculture analyses showed severe reductions in Teladosagia circumcincta, Trichostrongylus colubriformis as well as Haemonchus contortus, on days 7 to 28 for both studies down to zero. There are few reports in the literature on the use of plants to control Eimeria coccidia, all of them containing tannin such as Melia azedarach or Pistacia lentiscus, besides Sericea lespedeza, which has been effective on fresh green grazing or when fed dry as hay, leaf meal or pellets. The authors concluded that lespedeza provides a valuable environmentally friendly tool for improving the health and productivity of young goats in contrast to the chemical anthelmintic drug treatments with their build-up of resistance in the parasites and the danger of residues in meat and milk of treated animals. Lespedeza has great potential for both the prevention and treatment of coccidiosis or mixed infections of Eimeria and nematodes in young goats.

## 4. Different Goat Breeds

Goats come in more genetic varieties than cattle or even sheep [21]. There are two types of goats for two kinds of fiber, Mohair fiber from the Angora goats and Cashmere wool from Cashmere goats. Then, there are the genetically different meat goats like the heavy muscled Boer and the Savanna from South Africa; furthermore there are the "common" goats, which have become very profitable as brush clearers for fire break areas and for the reduction in obnoxious weeds like Kudzu, multiflora rose and creosote bush, besides being the most non-descript and most frequently kept to sustain small households, mountain farms and still provide also cash from sale of animals for breeding and for butcher [22].

Of course, last not least, there are the dairy breeds of goats, especially the Swiss varieties of Saanen, Alpine, Toggenburg, Oberhasli, and those from other parts of the world like the Nubian from the United Kingdom (U.K.), the Jamnapari from India, the Manchego sheep from Spain, the LaMancha from USA, and the Dwarf goats from Nigeria. The dairy breeds have been the more prominent and, over the years, the most persistently profitable goats, as they provide meat, and principally milk, not only for a majority of people around the world, but locally, in greater quantities than is needed by a small household [23]. A real dairy industry has developed in the last 100 years transforming goat milk into sales on the gourmet and health-conscious market like in Europe and USA, and by mail-transforming goat milk into many artisanal cheeses of many varieties and ages, and also yogurt, butter and cosmetic soap products, and even feeding goat milk to calves for prime-meat-quality veal calves. In Saanen goat-milking, the portable converted cow-milking machines made it easy milking for a small dairy goat farm operation as well as even university students in Delaware, while the disposal of large amounts of milk was more of a challenge [24]. However, instead of making cheese, as many new US goat farmers today are doing, feeding most of the milk at the rate of as much as 15-L per day of

fresh milk to each young Holstein bull calf proved to be a very profitable business, as the calves were easily gaining 2.5 kg bodyweight per day and were developing into prime meat class veal for sale [24].

## 5. Management

In a recent study of goats on 137 smallholder farms, keeping between 6 and 20 goats of the traditional non-milking type, in Malawi, East Africa [25], it was found that these goats ranked highest among the different livestock species for household income and resiliency to climate problems. Goats accounted for 61.2% for total livestock household income, while cattle, pigs and chickens contributed 17.6, 15.5 and 4.1%, respectively. Selling live goats constituted a major portion (79.2%) of the income, indicating that keeping goats was primarily for generating cash income. Thus, the average return on invested capital was 24.6%, exceeding the average commercial rate of 8% several fold.

Increasing numbers of intestinal parasite infections in small ruminant production systems around the world has become an increasing concern due to the widespread development of resistance of helminths to pharmaceutical treatments [26]. In a recent study of 350 sheep from five farms, commercial anthelmintic drugs were evaluated as to their efficacy in reducing fecal egg counts [27]. While all drugs were effective, each showed resistance to the drugs to varying degrees: Albendazole 33%, Levamisole hydrochloride 25%, Ivermectin 48%, Moxidectin 14%, Closantel 23%, and Ivermectin + Levamisole + Albendasole combination 30%. Thus, all drugs showed some degree of drug resistance, with Moxidectin showing the least. The impact of these resistances is worrying to the small ruminant industry, as helminths are becoming tolerant quickly, especially where drugs are used indiscriminately, even for new drugs, and the consequences are weight loss of the animals, decreasing milk, meat and wool yield, reduced reproduction and even death. This study called for more effective use of treatments, better rotation of pastures to eliminate re-infestation from feces on the ground, use of antihelminthic plants and their extracts, and genetic immunity of the animals.

Different from the traditional keeping of individual goats freely grazing around the countryside and mountains during the day and returning to the farm for the night, many times goats are instead found in flocks with sheep, and maybe even some cattle, to reduce brush encroachment on pastures under the guidance of a shepherd and his dogs to guard against predators and also keep small donkeys with the flock, with the same guarding task, of which they are very capable and effective. The shepherds would deliver the sheep back to their individual households for milking in the evening and for overnight security or, in some countries, one could find the shepherds living in a mobile trailer, keeping their flock in a mobile overnight coral in the transhumance system [28] covering mountain meadows, like in the Iberian Peninsula, the Alps or the Balkans, for several months before retiring for winter back into the village [29]. Under these systems, a problem of helminthiasis, internal parasitic worms, is often minimal or not considered at all in sheep, especially where the sheep are moved every day to new areas and are not exposed to grazing where they were yesterday, and where they had been excreting worm eggs, which then would be picked up the next day by the foraging sheep.

In more intensive systems of management, where goats are not allowed to roam freely on the open country or even along highways, they are sent out in flocks together in the morning to designated pastures next to the farms, but with limited and fenced-in areas of land, or they hike up the mountains, as in parts of Switzerland. Then, they return in the evening back to their home barn for milking and overnight rest. Preferably, these pastures are rotated daily, or at least weekly, for a flock of goats, to avoid re-exposure of the goats to their fecal egg excretions [10]. Nevertheless, under such systems, goats have varying helminth intestinal loads, which require medical treatment sooner or later, depending on the nutritional status of the goats, while well-fed goats often suffer less and may require less frequent treatments. Nevertheless, a recent study [30] of 123 mountain flocks grazing on communal summer pastures of Tyrol, Italy, revealed more than 90% applied anthelmintic treatments once or twice a year, although only 16% of the sheep and 30% of the goat farmers ever did fecal egg count examinations. Of the 1417 goats >12 months old, there were 23% negative, and of 1327 adult sheep, there were 16% negative in fecal egg counts, while kids and lambs had higher counts. The goat

flocks ranged from 5 to 125, with an average of 31, and the sheep flocks from 2 to 100, with an average of 28, although 29% had more than 40 animals. The breeds were mostly local and Alpine goats, and half of the goat farms were producing goat milk.

Gastrointestinal nematode infections have led to severe economic stresses in goat and sheep production systems and caused the search for different means of control than the series of pharmaceutical drug treatments, which have already developed growing resistance to the chemicals in the treated helminths against goats and sheep. The genetic selection of resistant breeds of small ruminants is considered [31], especially since heritability values of up to 0.65 of different animal breeds against fecal egg counts have been reported in previous studies. The Santa Ines sheep are considered to be highly resistant to internal parasites. They were compared with Dorper, Texel, Isle de France, crosses of 1/2 Ile de France and Dorper and crosses of 1/2 Texel and Santa Ines sheep for a total of 134 ewes, and the results showed that the 1/2 Texel and Santa Ines sheep had the significantly lower mean fecal counts (1.35 log 10), while the Dorper sheep had the highest (2.51 log 10), although the Isle de France had by far the lowest fecal egg count repeatability of 0.09, compared to 0.38 for the 1/2 Texel and Santa Ines sheep.

Continuing the search for alternative treatment methods to the chemical drug procedures and genetic selection, the use of a vaccine against the most important helminth Haemonchus contortus was studied [32] in Brazil, with 20 Saanen and 20 Anglo Nubian goats, because there is also the added concern of drug residue in the milk of goats and sheep. The vaccination regime started with goats of 6 months of age and continued for 511 days. They all produced fewer parasite eggs, 65% for the Nubians and 68% for the Saanen, throughout pregnancy and lactation. The vaccine did not affect milk yield, somatic cell count and general milk composition, and there was no withdrawal requirement for the milk, which is important in comparison to the use of chemical drugs. On microbiological examination, there was a significant reduction in mastitis-causing microorganisms of the genus coagulase-negative Staphylococcus.

Other genetic studies have extended the strategy of trying to eradicate gastrointestinal helminths to finding chromosome regions that control resistance mechanisms in the affected animals [33], specifically against Haemonchus contortus in Guadeloupe, with 182 Creole goats. Since infection by this helminth causes blood loss and inflammatory responses, it was expected to find genes responsible for these and related reactions thanks to the advances in molecular genetics and the identification of nucleotide polymorphism markers. Actually, seven locations on four chromosomes—4, 6, 11 and 16—were found to be associated with fecal egg counts that were part of gastrointestinal infections, intestinal damage, inflammation, immune response, hemorrhage and muscle weakness. Thus, genetic resistance strategies could become a novel approach to helminth control.

## 6. Eliminating Fecal Egg Counts

In the search for a successful fighting tool and procedure against the problem of internal parasites, we have an arsenal of pharmaceutical drugs, which are becoming decreasingly effective as time proceeds, with the development of genetic resistance of the parasites against any drug. Next comes the selection for the genetic resistance of certain goat and sheep breeds, and finally the use of vaccines. All these three procedures are fighting helminths to varying degrees without actually eliminating them. This, however, can be done with a more radical approach of the Integrated Crop-Livestock System [34], as practiced in Brazil, and which consisted of plowing under the parasite-infested pastures that used to be grazed by sheep, and planting corn and pigeon peas. After harvesting for silage, which is an interesting alternative farm activity. Including nitrogen fixation in the soil, the pastures were seeded with black oats by broadcasting, which was found to be "clean" 3 months later, and lambs were brought back for rotational grazing. All lambs were free of internal parasites. The use of this Crop-Livestock System resulted in declining degrees of gastrointestinal infections and satisfactory sheep performance. The benefits of this infection prophylaxis were of great importance, as it also reduced the costs of anthelmintic treatments and anthelminthic resistance.

Even more confined are dairy goats of the genetically higher-producing breeds, like Saanen, Alpine, Nubian, Oberhasli, LaMancha, yielding as much as 2–3 L milk per day, and they are not inclined, with their big udders, to do a lot of walking around the countryside. These goats may be kept in a dry lot system, as is found in the dry areas of California, or in straw-bedded pens inside buildings, as found in France in particular, where the goats are then fed silage as well as hay and commercial grain rations [29]. However, goats are only successfully kept separate from their feces and from fecal re-infestation when they are physically separated from their feces excretion. This is achieved by keeping goats confined in elevated housing with varying kinds of slatted floors, as practiced in Taiwan, with an entire flock of 500,000 dairy goats, mostly of the Alpine breed, that makes up a very successful goat fluid milk industry [35]. The goats are kept in separated groups and sorted into big houses by age, with maybe several hundred per pen. For milking, they move into adjoining elevated housing also with slatted floors, and they have an exit ramp onto an outside exercise lot, which they enjoy using. The housing has at least 2-m high clearance underneath for a person to collect the pellet goat feces, which fall through the slatted floors, and are sold for side income to home gardeners and stores. There may even be slanted sheets underneath so that the feces fall onto a middle alley for easier collection. At the Central Dairy Goat Research Station at Kaitung, mechanical scraper equipment was developed for easier feces collection. The milk is collected by a tank truck and processed at the Central Goat Milk Cooperative into bottled milk, even ultra-high temperature (UHT) milk, in 6-packs for home delivery. It is interesting to note that the Taiwanese believe in the superior health values of goat milk based on ancient Chinese tradition. This is in contrast to the equally modern goat milk industry in France and other countries, where goat milk is processed mostly into a variety of specialty cheeses, but where the goats are kept in bedded pens on the ground. Their fight against infections with gastrointestinal parasites then depends on the three different strategies discussed above to overcome the drug resistance of the parasites or utilize genetic the resistance of the host animals, while no elimination of fecal contamination is practiced. Outside of Taiwan, one can find elevated goat housing with slatted floors, wooden or expanded metal in the tropics, as goats prefer lying on the slatted floors on hot days, and because it is warm during cold winter days with snow on the ground. It is interesting to note that slatted floors are quite a common part of housing in Sweden, and the vast majority of Norwegian sheep barns are built with slatted floors [36]. Even earlier than these studies, the elevated goat housing with slatted floors was built in a dairy goat farm in Delaware and the picture was demonstrated in the Extension Goat Handbook [21]. As shown in Figure 1, the goats like to walk and rest on the slatted floor, also eating hay from a front manger, while the feces are falling through the separated slats to the floor for separation from the goats. No straw bedding is needed, and no anthelminthic drugs were ever used. Certainly, a wider acceptance of slatted-floor elevated housing would be very desirable and profitable. The effects of slatted floors and manure scraper systems on the concentrations and emission rates of ammonia, methane and carbon dioxide in goat buildings have been also reported recently [37].

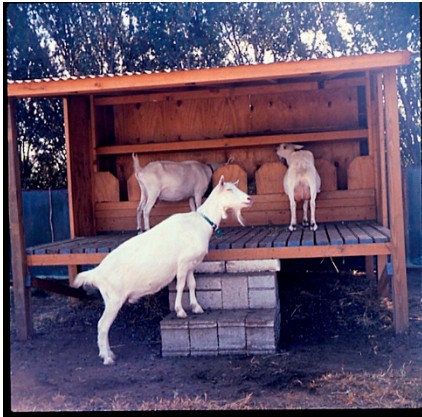

**Figure 1.** Delaware model of elevated slatted floor housing [21].

## 7. Conclusions

Helminthiasis, or gastrointestinal infection with nematode parasites, is an increasing management problem in the sheep and goat industry, causing weight losses, diarrhea, poor growth, decreased milk production, reproductive failure, and eventually death. Increasing resistance to drug treatments is causing infectiveness in drug treatments and the need to search for new ways to prevent and treat animals more effectively. As an alternative to different applications of different new or combined drugs, there are indications of different breeds of sheep and goats that are genetically immune or are able to acquire immunity. Immunisation with vaccines has shown promise. Traditional herbal medicine with plants which contain nematodicidal compounds like tannin have shown great potential in reducing gastrointestinal infections and providing safe forage feeding. The legume Sericea lespedeza, in particular, has proven in a number of extensive studies to be worthy of more widespread forage use for goats, while its tannin contents make it not acceptable for sheep feeding. A third approach to controlling helminthiasis is eliminating goat and sheep feces from the grazing areas, which is practiced under the traditional transhumance system nomadic shepherding or by plowing certain areas that were grazed and will be grazed again after one or two years of crop production with corn or soybeans. Finally, feces can be prevented from reinfesting grazing goats not only by the rotational use of different pasture lots, but by managing goats in total confinement housing, which is elevated and has slatted floors. This system of slatted floors is totally practiced by the entire dairy goat population in Taiwan very successfully, and deserves wider attention in the milk and meat goat industry around the world. Interestingly, slatted flooring is already widely found in the sheep and goat industries in Sweden, Norway, Malaysia and Guatemala.

**Author Contributions:** Writing-Original Draft Preparation: G.F.W.H. and Y.W.P.; Writing-Review & Editing: G.F.W.H. and Y.W.P. All authors have read and agreed to the published version of the manuscript.

**Funding:** This research was partially funded by the grants of NIFA/USDA Evans-Allen funds of GEOX-3225.

**Conflicts of Interest:** The authors declare no conflict of interest.

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
