# Peer review of "Fighting the Deadly Helminthiasis without Drug Resistance"

_2624-862X, doi:10.3390/dairy1030012_

Round 1
Reviewer 1 Report
If the authors try to collect more recent references used, the value of the review will be higher than now, because of the references of the issues of the 1990s. And, its tile is absolutely too exaggerated, since the content is focused on JUST sheep and goats. If they do not mind that I recommend " Fighting the severe helminthiasis of small ruminants without drug resistance" or something like that.
Author Response
Reviewer #1:
“ Extensive editing of English language and style required ”
<Ans> The senior author of this article has served as an Editor-in-chief for more than 20 years for a traditional major journal, so that we think the extensive editing of English language and style may not be needed.
“If the authors try to collect more recent references used, the value of the review will be higher than now, because of the references of the issues of the 1990s.”
<Ans> The reviewer has a good point on this premise. However, we have cited some recent research reports, which would demonstrate the essence of the condensed tannin forages such as Sericea Lespedeza in inhibiting GIN nematodes and reducing internal parasite egg counts in ruminants. More citations of recent articles may strengthen the review, while the extreme constrains of our time limited the task. However, we believe that the article contains the essentially needed information, whereby more recent reports may not be necessary.
“And, its tile is absolutely too exaggerated, since the content is focused on JUST sheep and goats. If they do not mind that I recommend " Fighting the severe helminthiasis of small ruminants without drug resistance" or something like that.”
<Ans> Although the reviewer #1 has a good suggestions on these remarks, we respectfully do not agree with these comments. Even if the manuscript describes mainly on sheep and goats, it is inclusive for all ruminant species. For example, lines 106-107 and other places talk about cattle, sheep and goats, indicating that the treatment of anthelmintic tannin containing forages can be applied to cattle and other ruminant species. In addition, in our view, changing the title of our article by the wordings of the reviewer’s suggestion, would not make much difference from the original title, so that we have decided to leave the title of the paper as its original form.
Reviewer 2 Report
It is an interesting paper that reviews one of the main concerns in small ruminants farms today. Although there are previous several specific reviews about anthelmintic resistance (AR) authors have revised the problematic and propose alternatives to avoid the consequences of AR. This alternative tools may contribute to delay the development of AR and mitigate its consequences.
Formal questions
Although they are usually known by the readers, expressions as Monounsaturated Fatty Acids (MUFA) (line 99) should be complete written the first time that they are used.
similar for VFA (line 113) UHT (line 299)..
Genera and species should be written in italics
Perhaps UK is better than U.K. (line 170) Similar to USA
Small questions:
Line 119/120/table 1 commercial supplement: Main compound/s?
Line 124 Table 1. Nematode fecal egg count
Line 157 Could be GIN better that just nematode?
Line 170 There is no a Manchego goat in Spain (there is a Manchego sheep) Do you refers to Murciano-Granadina goat?
Line 171 Dwarf (all the rest are written in capital letter)
Line 190 17.6, 15.5 and 4.1%, respectively.
Lines 211 to 219 Goat flocks do not really make transhumance in the Iberian Peninsule. By contrast, it is very common the sheep flock transhumance (and in many of these sheep flocks there is a very small number of goats) Please, consider it in your arguments that could be good in other countries, as you mention, but in Iberian Peninsule may be slightly different
Line 332 and following:
Helminthiasis includes also other helminths, not only GIN
“..Increasing management problem of the sheep and goat industry causing weight losses, diarrhea, poor growth, decreased milk production, reproductive failure and eventually death because increasing resistance….” In my opinion it is not completely true. These clinical signs may occur when there is drug resistant nematodes in the flocks but they also can occur when nematode population are susceptible. The real problem of resistance is as “…increasing resistance to drug treatments is causing infectiveness of drug treatments and the need to search for new ways to prevent and treat animals more ….” I think that changing “because” in line 324 for a point separating two sentences will be more realistic. I think that it is not demonstrated that resistant nematodes be more pathogenic that susceptible strains.
Author Response
“I don’t feel qualified to judge about the English language and style”
<Ans> As we described for this issue for Reviewer #1, we feel the English would be okay.
“It is an interesting paper that reviews one of the main concerns in small ruminants farms today. Although there are previous several specific reviews about anthelmintic resistance (AR) authors have revised the problematic and propose alternatives to avoid the consequences of AR. This alternative tools may contribute to delay the development of AR and mitigate its consequences.”
<Ans> Yes, we agree with the Reviewer #2’s comments and remarks in general comments. By using alternative tools through feeding anti-parasitic tannin containing forages such as Sericea Lespedeza, or using elevated housing with slated floors would mitigate the severe helminthiasis especially in small ruminants.
Formal questions:
“Although they are usually known by the readers, expressions as Monounsaturated Fatty Acids (MUFA) (line 99) should be complete written the first time that they are used.”
<Ans> We inserted the whole spellings in line 96. It is not in line 99.
“similar for VFA (line 113) UHT (line 299)..
Genera and species should be written in italics
Perhaps UK is better than U.K. (line 170) Similar to USA”
<Ans> We inserted all the words for the abbreviations in the revised manuscript, except USA, which is too common and everyone knows about it.
Small questions:
“Line 119/120/table 1 commercial supplement: Main compound/s?”
<Ans> The main compositions of commercial supplement were composed of
cracked corn, cottonseed meal, liquid molasses, which were made as pellets.
“Line 124….Table 1. Nematode fecal egg count.”
<Ans> It is a good suggestion, but we cannot change the cited original table content.
“Line 157……There is no a Manchego goat in Spain………….Murciano-Granadina goat?”
<Ans> It is an excellent comment for correcting errors in this regard. Yes, the Reviewer #2 is correct in these comments. Manchego sheep is correct, so that it has been corrected in the text.
“Line 171……..Dwarf…………….”
<Ans> It has been corrected as capital letter.
“Line 190………….. 17.6, 15.5 and 4.1%, respectively.”
<Ans> The word “respectively” has been added in the sentence.
“Lines 211 to 219. Goat flocks do not really make transhumance in the Iberian Peninsule. By contrast, it is very common the sheep flock transhumance (and in many of these sheep flocks there is a very small number of goats) Please, consider it in your arguments that could be good in other countries, as you mention, but in Iberian Peninsule may be slightly different.”
<Ans> This is very excellent observation by the reviewer #2, where the reviewer is correct that sheep flocks make transhumance in the Iberian Peninsule. We have corrected this important errors found by the Reviewer #2. We have corrected this error in the lines 211-219 for the revised version.
“Line 332 and following: Helminthiasis………….not only GIN”…………….
“Increasing management problem of the sheep and goat industry causing weight losses, diarrhea, poor growth, decreased milk production, reproductive failure and eventually death because increasing resistance….” In my opinion it is not completely true. These clinical signs may occur when there is drug resistant nematodes in the flocks but they also can occur when nematode population are susceptible. The real problem of resistance is as “…increasing resistance to drug treatments is causing infectiveness of drug treatments and the need to search for new ways to prevent and treat animals more ….” I think that changing “because” in line 324 for a point separating two sentences will be more realistic. I think that it is not demonstrated that resistant nematodes be more pathogenic that susceptible strains.”
<Ans> We accepted the reviewer #2’s suggestion on the sentence, where “because” in line 324 has been removed, and made the previous one sentence into two separate sentences as the reviewer requested. This may make more sense, as the reviewer suggested.
We greatly appreciated both reviewers for their contributions to the revision of our article. We hope our revisions would be acceptable to the two reviewers and you.
If you have any further comments and suggestions, please let us know.